# Subdividing Stress Groups into Eustress and Distress Groups Using Laterality Index Calculated from Brain Hemodynamic Response

**DOI:** 10.3390/bios12010033

**Published:** 2022-01-09

**Authors:** SuJin Bak, Jaeyoung Shin, Jichai Jeong

**Affiliations:** 1Department of Brain and Cognitive Engineering, Korea University, Seoul 02841, Korea; soojin7897@korea.ac.kr; 2Department of Electronic Engineering, Wonkwang University, Iksan 54538, Korea; jyshin34@wku.ac.kr

**Keywords:** functional near-infrared spectroscopy, international affective picture system, laterality index, saliva alpha-amylase, stimulus-response tasks, stress measurement

## Abstract

A stress group should be subdivided into eustress (low-stress) and distress (high-stress) groups to better evaluate personal cognitive abilities and mental/physical health. However, it is challenging because of the inconsistent pattern in brain activation. We aimed to ascertain the necessity of subdividing the stress groups. The stress group was screened by salivary alpha-amylase (sAA) and then, the brain’s hemodynamic reactions were measured by functional near-infrared spectroscopy (fNIRS) based on the near-infrared biosensor. We compared the two stress subgroups categorized by sAA using a newly designed emotional stimulus-response paradigm with an international affective picture system (IAPS) to enhance hemodynamic signals induced by the target effect. We calculated the laterality index for stress (LIS) from the measured signals to identify the dominantly activated cortex in both the subgroups. Both the stress groups exhibited brain activity in the right frontal cortex. Specifically, the eustress group exhibited the largest brain activity, whereas the distress group exhibited recessive brain activity, regardless of positive or negative stimuli. LIS values were larger in the order of the eustress, control, and distress groups; this indicates that the stress group can be divided into eustress and distress groups. We built a foundation for subdividing stress groups into eustress and distress groups using fNIRS.

## 1. Introduction

Stress is rapidly becoming one of the most common mental disorders in modern society. The number of studies on stress, including diagnosis, detection, and rehabilitation, has gradually increased and thus attracted attention from many researchers [1]. Brain neurologists focusing on mental stress follow stress protocols to obtain distinct brain images. Traditional stress protocols are used to evaluate stress precisely [2,3]. A stimuli-response task is a representative stress protocol in which one responds to warning stimuli (WS) followed by target stimuli (TS) using emotional international affective picture system (IAPS) images [4]. The process of finding positive or negative images of TS not only increases people’s attention but also elicits momentary mental stress responses. Some studies have often reported high amplitudes of positive or negative hemodynamic signals in stressed people using target images [5,6]. This phenomenon is known as the target effect [5]. Specifically, the target effect refers to the increase in the intensity of biosignals while looking for intermittently given TS. The target effect has already been fully demonstrated in many neuroimaging studies [7,8]. 

Functional near-infrared spectroscopy (fNIRS) is a state-of-the-art optical neuroimaging technique based on near-infrared biosensors that are used to identify stress levels. Physically, fNIRS has the advantage of being more accessible and convenient than other devices [9,10,11,12]. Additionally, brain hemodynamic responses acquired by fNIRS not only exhibit apparent differences between stressful tasks and rest states but also facilitate stress detection and diagnosis [13,14,15]. By taking advantage of these characteristics, the stress level can be easily quantified by the concentration changes of oxygenated hemoglobin (ΔHbO), reduced oxygenated hemoglobin (ΔHbR), and total hemoglobin (ΔHbT) from hemodynamic oscillations [16].

Despite the existence of neuroimaging technologies to quantify stress, evaluating stress still remains a challenge because of the lack of evidence for brain hemodynamic activities in stressful people. Moreover, it is difficult to divide stress into two different categories: positive-type stress, named as “eustress,” which has a low-stress level/a high behavioral result, and negative-type stress, called “distress,” which has a high-stress level/a relatively low behavioral result [17]. According to the inverted U-shaped Yerkes–Dodson model [18], eustress can yield high task accuracy and fast reaction times, whereas distress can not do so.

Despite the opposite characteristics of both types of stresses, most stress-related neuroimaging studies have not distinguished between eustress and distress [19,20]. However, some studies have recently attempted to subdivide stress. One study classified common stress and eustress by heart rate and survey data, showing an accuracy of up to 71% using correlation and principal component analysis [21]. Another study found the brain’s electrical activity in the frontal and central lobes in identifying eustress and distress [22], but this was not proven.

Biomarkers, such as salivary alpha-amylase (sAA), are widely used and have been fully proven in the field compared to neuroimaging [23,24]. Lim et al. [25] demonstrated that it is possible to distinguish between eustress and distress by sAA levels because the sAA level rapidly increases while participants perform a stress-inducing task [26]. Thus, we need to use the already validated sAA as a reference and then compare it with the cerebral blood flow oscillations between eustress and distress using neuroimaging techniques.

In many neuroimaging studies, the laterality index (LI) is defined as a hemispheric dominance in mental stress-induced cognitive tasks, such as mental arithmetic, working memory, etc. [27]. The laterality index for stress (LIS) is mainly used to calculate an asymmetry coefficient in left/right brain patterns between stressed people and unstressed people [28]. Most studies have shown dominant brain activity in the right hemisphere of people stressed by mental stress tasks [29,30,31]. Here, the right hemisphere plays an important role in the diagnosis of stress. Furthermore, LI-based studies have reported that high- and low-stress can be evaluated in difficult and easy-task stages, respectively; The more difficult the task to be performed is, the greater the perceived stress and the dominant activities appear in the right prefrontal cortex (PFC) by LI calculation [32,33]. However, it has never been reported that the perceived stress is directly related to the physiological stress measured objectively, such as sAA. Moreover, the stress measured by sAA is not yet known whether hemisphere-specific LIS changes influence the PFC in the eustress and distress groups [34]. To address these issues, we divided the actual stress measured by sAA into eustress (low stress) and distress (high stress) and then calculated the LIS values by the hemodynamic oscillations of fNIRS. Furthermore, our research can greatly contribute to measuring the stress lateralization between eustress and distress and differentiate stress groups in detail. 

Therefore, this study aims to distinguish between eustress and distress types by using LIS values calculated from fNIRS. The existing emotional stimulus-response paradigm was designed to enhance hemodynamic signals by the target effect and hemodynamic oscillations that induce positive or negative emotions visually. During the implementation of this paradigm, continuous hemodynamic oscillations with different stress levels in the PFC were measured and analyzed. Stress questionnaire scores were obtained, sAA levels were determined, and biosignals were recorded for comparison. The LIS values of stimulation-related hemodynamic oscillations were calculated; the stress group was subdivided into eustress and distress groups using the LIS value, which is consistent with the sAA test results. This study provides, for the first time, that the LIS values obtained from hemodynamic oscillations of fNIRS can be used to subdivide the stress group into eustress and distress groups.

## 2. Materials and Methods

### 2.1. Participants

We acquired sAA data from the participants for 1 min using a COCORO METER (Nipro, Osaka, Japan), which served as a physiological stress scale before the experiment. Apart from participants in the middle of their menstrual cycles (female), 44 right-handed task-naïve participants with normal or corrected-to-normal vision participated in the experiment. The participants were asked not to consume caffeine or food and not to exercise for at least 2 h before the experiment [35]. Written informed consent was obtained from all participants, and they completed the anxiety and stress questionnaires before starting the experiment. Blood pressure (BP) and respiratory rate were measured. A control group of 22 participants (15 males, aged 24–27 years, sAA level < 28 KU/L) and a stress group of 22 participants (15 males, aged 24–27 years, sAA level > 33 KU/L) based on sAA levels took part in the experiment. The stress group was subdivided into the eustress group (sAA level < 44 KU/L) and distress group (sAA level > 44 KU/L). Eleven participants per group (control, eustress, and distress) were randomly selected to create a balanced sample set for statistical validation. Statistical analyzes were repeated thrice using randomly chosen subsets of the control group to prevent biased results, thereby leading to similar statistical outcomes. Table 1 summarizes the participants of each group.

### 2.2. Apparatus

The fNIRS data were recorded using a two-wavelength (780 and 850 nm) continuous-time multichannel fNIRS system (NIRSIT LITE, OBELAB, Seoul, Korea) consisting of 15 channels. Figure 1 presents the placement of the fNIRS optodes. Five light sources and seven optical detectors were placed on each participant’s forehead, covering the PFC. The adjacent source-detector was at a distance of 25 mm. The light intensity data were collected at a sampling rate of 8.138 Hz. Data were converted to changes in the concentration of oxygenated and reduced hemoglobin (ΔHbO/R) using a built-in function based on the modified Beer–Lambert law [36]. The acquired fNIRS dataset, and all related information are downloadable. Available online: https://github.com/SujinBak/SubdividingStress (accessed on 16 November 2021).

### 2.3. Experimental Paradigm

The experimental process began with the participants completing the anxiety and stress questionnaires, followed by measuring the physiological signals of participants, including sAA levels. Then, the participants performed the experimental paradigm.

The experiment was conducted between 12:00 and 4:30 p.m. to avoid the influence of the circadian pattern on cortisol [37,38] under good air conditions, and the illumination ranged from 900.8 to 917.9 lx (Light meter, UYIGAO, Shenzhen, China) to avoid any environmental stress. The participants were seated on a chair in front of a display monitor (full HD LED 27-inch, FLATRON IPS277L-PN, LG, Korea, Nanjing). The display resolution was 3440 × 1440 with a refresh rate of 60 Hz. Visual stimuli were presented at a visual angle of 7.6° and 75 cm away from the participant. Participants were also asked to minimize head movement and remain calm throughout the experiment.

The experimental paradigm was implemented using a Matlab^®^ App Designer and was performed in two different sessions, counterbalancing among participants. We designed an experimental paradigm to reveal brain patterns of stress groups sensitive to negative images (Session 2) and compared them with positive images (Session 1). The experimental paradigm is illustrated in Figure 2. Each session lasted approximately 20 min and consisted of 20 repetitions of trial number display (1 s), stimulus-response task (5 s), and inter-task break (35 s) periods. A single stimulus-response task (trial) was composed of five repetitions of a warning stimulation display (0.5 s) with a short beep (50.3 dBA, sound level meter, YATO, China), an anticipatory period without any display (2 s), both the TS display and the participants’ response (2 s), and feedback displayed (0.5 s) with different aural alarms (correct answer: 42.8 dBA, incorrect answer: 42.4 dBA, time out: 40.6 dBA). 

To utilize the target effect during fNIRS measurements [5], in Session 1, a smiling face was shown for the warning stimuli (WS) period, and then positive or negative images were randomly shown for the TS period. The percentage of positive images was 30%. However, in Session 2, an angry face was shown for the WS period, and then negative or positive images were randomly shown for the TS period. The ratio of negative images was 30%. When the TS was matched to the WS, the participants were instructed to press the right-arrow key on a computer keyboard as quickly as possible to mark the task accuracy and response time.

### 2.4. Valence (from Negative to Positive Emotions) and Arousal (from Calm to Excitement) Ratings

Russell’s Circumplex Model [39] provides two main dimensions (affective valence and arousal) to classify emotional stimuli. Here, valence was represented from negative to positive, and arousal was represented from calm to excitement. It is necessary to divide the international affective picture system (IAPS) images into valence and arousal to induce positive or negative emotions in participants [40]. Based on this model, 100 positive and 100 negative images were chosen from the IAPS image dataset. Table 2 presents the selected image numbers that indicate the visual stimulus ratings. Positive images indicate (mean value [μ] ± standard deviation [σ]: 7.47 ± 1.48) valence rate with a (4.8 ± 2.31) arousal rate, whereas negative images indicate a (2.22 ± 1.44) valence rate with a (5.96 ± 2.20) arousal rate.

### 2.5. Signal Processing

We adopted the signal processing methods used by Al-Shargie, F., et al. [41,42]. Using the modified Beer–Lambert law, changes in oxygenated and reduced oxygenated hemoglobin concentrations converted from fNIRS signals can be represented as ΔHbO and ΔHbR, respectively. The converted signals were preprocessed to remove low-frequency drift and high-frequency system noise using the framework of the BBCI-Toolbox [43] running in Matlab 2019a (MathWorks, Natick, MA, USA). The signals were digitally bandpass-filtered through a zero-phase filter implemented by a sixth-order Chebyshev filter with a passband of 0.01–0.02 Hz (approximately 1/60 s) to remove physiological noise and DC offset. We obtained the same results using a wide passband of 0.01–0.1 Hz to prevent information loss due to the narrow passband of 0.01–0.02 Hz. The filtered hemodynamic responses were segmented into epochs ranging from −1 to 60 s relative to the task onset (i.e., 0 s). Epochs were subjected to a baseline correction, which consisted of subtracting the average value within the reference interval ranging from −1 to 0 s. The temporal means of ΔHbO in each channel were calculated by averaging the fNIRS data from the onset (0 s) to the termination time (60 s) of each epoch. Similarly, ΔHbR was also preprocessed, but ΔHbO signals better reflect the hemodynamic activation [44]. Thus, we dealt only with ΔHbO signals in the subsequent analyzes [45].

### 2.6. Anxiety and Stress Scales

All questionnaires were translated into Korean and completed before the physiological measurements were taken.

#### 2.6.1. State-Trait Anxiety Inventory (STAI)

We instructed the participants to carry out the State-Trait Anxiety Inventory (STAI); a universal scale used to measure psychological inventory. Although some incidents or situations cause anxiety, the STAI-Trait (STAI-T) is used to assess chronic anxiety. Furthermore, STAI-State (STAI-S) is used to determine the acute state of tension and nervousness caused by the threat of an event or situation. Both the STAI-T and STAI-S consist of a self-reported assessment of 20 items that are to be evaluated using a 4-point Likert scale. For the STAI-T, 54–58 points indicate slightly higher trait anxiety, and 59–63 points indicate relatively high trait anxiety. At ≥64 points, the trait anxiety was considered very high. A higher score indicates a higher level of trait anxiety. In the STAI-S, 52–56 points indicate slightly higher state anxiety, and 57–61 points indicate relatively high state anxiety. At ≥64 points, the state anxiety is considered very high. The higher the score, the worse the state of anxiety.

#### 2.6.2. Perceived Stress Scale 

Perceived Stress Scale (PSS) is a criterion that focuses on how participants perceive and interpret stress over the past month based on a 5-point Likert scale. It is suitable for assessing the level of stress that an individual feels because it can show a difference in the stress levels of each individual in the same stressful situation. The higher the score, the greater the perceived stress level.

### 2.7. Physiological Data Recording: Pressure (BP) and Breath per Minute (BPM)

To investigate physiological conditions or abilities among participants, we measured two data points, BP and BPM, before performing the stress task. The BP was measured in the right forearm with a BP gauge (BP A6 PC, Microlife, Taiwan, China) in the correct posture. Systolic blood pressure (SYS), diastolic blood pressure (DIA), and pulse rate were measured thrice to obtain precise averaged pulse rates and BPs based on the blood pressure standards of the World Health Organization [46]. Furthermore, BPM was measured with a respiration belt (Go Direct Respiration Belt, Vernier Software & Technology, Beaverton, OR, USA) worn under the chest on the left while breathing freely for a minute in a comfortable position.

### 2.8. Behavioral Measures

We measured the response accuracy ratio (i.e., the number of correct responses divided by the number of responses in each session) and response times (i.e., the elapsed time from the onset of stimulation to a participant’s response).

### 2.9. Laterality Index for Stress (LIS)

LI is commonly used to describe the asymmetry of brain activation [47]. We calculated the LIS for each session of each group by using the formula LI=(R−L)/(R+L), where R and L are the maximum absolute amplitudes of ΔHbO according to Reference [48]. There are channels on the left (Chs. 9–15) and right hemispheres (Chs. 1–7), respectively. LIS values ranged from −1 to 1; a positive value (0 to 1) indicated right lateralization, and a negative value (−1 to 0) indicated left dominance.

### 2.10. Statistical Analysis

Levene’s test was used to assess the homogeneity of variance in each group. Then, two independent sample *t*-tests and one-way analysis of variance (ANOVA) were applied to determine the differences between the two groups (i.e., control vs. stress) and between the three groups (i.e., control vs. eustress vs. distress). Dunnett T3 was conducted as a post hoc test based on the studentized maximum modulus [49]. We also investigated the correlations between sAA and the behavioral results of accuracy or reaction time. Hence, we obtained Pearson correlation coefficients for these data. 

## 3. Results

### 3.1. Behavioral Results

In the two-group analysis, the grand averaged response accuracy ratio of the stress group (μ ± σ: 96.3 ± 1.6%) was higher than that of the control group (95.6 ± 2.5%). The control group responded 4 ms faster in Session 1 (positive stimuli) than in Session 2 (negative stimuli), whereas the stress group responded 8 ms faster in Session 2 than in Session 1. In the three-group analysis, the grand averaged response accuracy ratios were highest in the eustress (96.9 ± 0.8%) group, followed by the distress (95.8 ± 2.0%) and control groups (95.5 ± 3.4%). The eustress and distress groups showed the fastest and slowest responses in both sessions, respectively. However, there were no statistically significant differences in accuracy and response times.

To reveal a correlation between stress levels (sAA) and behavioral results obtained from the subject’s key-pressing actions, we performed a two-tailed Pearson’s correlation analysis, as presented in Table 3. In the two-group analyzes, the highest positive correlation was observed between sAA level and accuracy (r = 0.484, p < 0.05 *). On the other hand, in the three-group analyzes, the highest positive correlation was observed between sAA level and response time in Session 1 (r = 0.429, p < 0.05 *). These results indicate that the accuracy and response time obtained from the task are related to the sAA, which provides another possibility to classify stress levels between eustress and distress groups.

### 3.2. Stress Scales and sAA

Table 4 presents the questionnaire scores and their statistical significance. The questionnaires (STAI-T/S, PSS) could significantly discriminate between the control and stress groups (STAI-T: t = 0.005, p < 0.01 **; STAI-S: t = 0.020, p < 0.05 *; PSS: t = 0.008, p < 0.01 **). However, the questionnaires were unable to distinguish between the three groups in the same order as above (STAI-T: t = 0.081, NS; STAI-S: t = 0.335, NS; PSS: t = 0.102, NS).

A two-tailed Pearson’s correlation analysis was performed to determine the relationship between sAA levels and all questionnaire outcomes for the two groups and among all three groups. The Pearson’s correlation coefficients were calculated. The highest positive correlation was observed between the sAA level and the STAI-T score for all groups (r = 0.346, p < 0.05 *). Conversely, the lowest positive correlation (r = 0.299, p < 0.05 *) was observed between sAA and PSS. Additionally, the correlation between the sAA level and STAI-S score was statistically significant (r = 0.310, p < 0.05 *). All questionnaires were directly proportional to the sAA levels. In the case of the three groups, the highest positive correlation was observed (r = 0.357, p < 0.05 *) between the sAA level and STAI-T score. The STAI-T questionnaire was directly proportional to the sAA levels. Conversely, the correlations between the sAA level and the STAI-S score (r = 0.307, NS) or PSS score (r = 0.305, NS) were not statistically significant.

### 3.3. Physiological Results

The physiological signals measured before the experiment were within the normal range. Data for four physiological parameters are provided in the order of SYS, DIA, pulse rate, and BPM in the following sentences. In the two groups, the measurements for the four physiological parameters of the control group were 117.4 ± 12.5 mmHg, 73.7 ± 9.0 mmHg, 78.2 ± 2.6 beats/min, and 19.2 ± 4.6 times/min on average. The stress group data were averaged to be 118.0 ± 13.1 mmHg, 73.0 ± 8.0 mmHg, 77.4 ± 10.8 beats/min, and 15.5 ± 4.7 times/min.

In the three-group comparison, the control group data were 111.5 ± 11.3 mmHg, 70.6 ± 6.0 mmHg, 75.0 ± 8.0 beats/min, and 19.7 ± 4.5 times/min. The eustress group data were 120.7 ± 13.0 mmHg, 74.3 ± 7.9 mmHg, 76.8 ± 12.0 beats/min, and 16.1 ± 15.1 times/min. Finally, the data of the distress group were 115.2 ± 2.6 mmHg, 71.3 ± 8.1 mmHg, 77.2 ± 9.8 beats/min, and 14.8 ± 4.2 times/min. The physiological results of all groups were observed under normal and similar conditions before the experiment.

Using *t*-test, statistically significant differences were not observed in any of the physiological data, except for BPM (t = 2.674, p < 0.05 *) between the control and stress groups, indicating that there is a significant difference in BPM between the two groups. We also found the highest negative correlation between the sAA level and BPM (r = 0.303, p < 0.05 *) using the two-tailed Pearson’s correlation analysis, as presented in Figure 3. 

### 3.4. Hemodynamic Responses: Two Groups

Figure 4a graphically illustrates the task-related ΔHbO for both the control and stress groups. In particular, the ΔHbO of the stress group generally showed a smaller amplitude (i.e., lower activation) than that of the control group. The task-related ΔHbR in Session 1, illustrated in Figure 4b, has an opposite sign for ΔHbO in all channels. The stress group showed a lower amplitude than that of the control group. Figure 5a presents the task-related ΔHbO in Session 2. The task-related ΔHbO of the stress group showed greater activation than that of the control group on all channels, excluding ch2 and ch3. Figure 5b presents the distinct task-related ΔHbR in Session 2. The ΔHbR of the stress group generally increased more than that of the control group.

### 3.5. Hemodynamic Responses: Three Groups

Figure 6 and Figure 7 depict the grand averaged ΔHbO and ΔHbR across all participants for the three groups obtained in Sessions 1 and 2, respectively. In all sessions, the highest peaks in ΔHbR and ΔHbO dips in ΔHbO were observed in the distress group, whereas the smallest amplitude of the hemodynamic oscillations was observed in the control group. Overall, the differences in hemodynamic oscillations were observed among the three groups.

### 3.6. LIS

Figure 8 presents the bar graphs of the averaged LIS between the two groups and among the three groups according to the experimental condition type of Session 1 (focusing on positive image) and 2 (focusing on negative image). A positive LIS value (0 to 1) indicates right lateralization, one of the stress symptoms caused by negative stimulation. According to Session 1 in Figure 8a, the LIS values of the stress and control groups were 0.02 and 0.12, respectively, but the stress and control groups obtained the same LIS value (0.06) in Session 2. This indicates that the positive LIS values of both the stress and control groups in Sessions 1 and 2 were relatively higher on average in the right prefrontal channels than in the opposite sides. We also discovered that the stress group was still affected by negative stimulation by showing larger LIS values in Session 2 than in Session 1. This result is consistent with that of a previous study [50]. Moreover, we obtained a large difference in the LIS values between the stress and control groups in Session 1, but not in Session 2. This means that the stress and control groups showed a distinctive difference in LIS values in Session 1.

According to Session 1 in Figure 8b, the LIS values of the eustress and distress groups were 0.197 and 0.004, respectively, but the LIS values were 0.320 and 0.004 for the eustress and distress group in Session 2. Similarly, the positive LIS values calculated from the eustress and distress groups are shown as right lateralization. More specifically, the properly stressed eustress group experienced dominant right frontal hemisphere activity, but we discovered for the first time that the over-stressed distress group exhibited little right-front hemispheric activity. These results imply that brain lateralization is not shown regardless of emotional stimuli because of cognitive decline caused by excessive stress in the distress group. Therefore, we obtained larger LIS values in the order of the eustress, control, and distress groups in both Sessions 1 and 2. As a result, this large difference in LIS values makes it possible to subdivide the stress group into the eustress and distress groups, even in Sessions 1 or 2. 

## 4. Discussion

Using fNIRS, our study was designed to discover the possibility of subdividing the stress group into subgroups (eustress and distress groups) that were consistent with using the sAA level as the threshold. To this end, we compared the LIS values in both subgroups, calculated from the hemodynamic responses of fNIRS. Furthermore, an experimental paradigm with two different types of visual stimuli that induce positive or negative emotions was suggested to elicit the best emotion that participants currently feel over time. The results of this study demonstrated differences between the groups, showing distinct LIS calculations from hemodynamic oscillations.

### 4.1. Differences in Behavioral Results

Stress is implied to have a negative effect on human performance measures, such as task accuracy and response time. However, the results of our study were contradictory. The task accuracy of the stress group was better than that of the control group in both Sessions 1 and 2. Moreover, in Session 2, the stress group responded more quickly than the control group. We speculate that this finding is rooted in the dominant influence of the characteristics of the eustress group on the overall stress group. Since it is generally acknowledged that the ability of performing tasks follows the inverted U-shaped Yerkes–Dodson model, the abilities of performing tasks under a positive stress condition (eustress) can be more enhanced than that under extreme conditions (negative stress called distress) [18,19,20,21,22,23,24,25,26,27,28,29,30,31,32,33,34,35,36,37,38,39,40,41,42,43,44,45,46,47,48,49,50,51]. This explicitly means that the members of the eustress group work faster and more efficiently than those of the other two groups. This study supports the findings of other studies [52,53]. 

Furthermore, we found a correlation between sAA and behavioral results, such as accuracy and reaction time (Table 3). In the two groups, there was a statistically significant correlation between sAA and accuracy. In the three groups, a statistically significant correlation was found between sAA and reaction time in Session 1. This implies that there is a change in the behavioral results depending on the stress level.

### 4.2. Differences in Questionnaire Results

Questionnaires have traditionally been used as indicators of stress before using sAA levels [54]. The results of the questionnaire in this study were consistent with those of previous studies [55,56], which indicate that the questionnaires could distinguish between the control and stress groups based on differences in the stress scale scores. It is notable that differences in stress scale scores were statistically significant between the control and stress groups, but not between the three groups (i.e., control vs. eustress vs. distress). This leads to the fact that a separation into three groups is no longer achievable with only the questionnaire scores. In other words, stress questionnaires cannot divide stress groups into eustress and distress groups.

### 4.3. Reduced ΔHbO Due to Stress States

Recent fNIRS-stress studies have shown that people with stress have reduced ΔHbO (negative peak) during cognitive stress tasks [57]. In other words, the hemodynamic signal (ΔHbO) has a high negative amplitude. This is consistent with our results and findings of the studies related to human and animal stress using functional magnetic resonance imaging (fMRI) because of reduced cortical activity in the PFC [58,59]. Moreover, Bryant et al. proposed using the ΔHbO as a biomarker to detect stress [60]. In our study, stress levels can be quantitatively detected with ΔHbO, which is in line with the results obtained by Aeschliman et al. [61]. This confirmed that stress induces a faster respiratory rate and hyperventilation; therefore, the stress group had a relatively higher amount of ΔHbO changes than the control group because the higher the stress level, the greater the hemodynamic changes. Thus, this study obtained distinct hemodynamic responses among the three groups by quantifying ΔHbO, as presented in Figure 6 and Figure 7.

### 4.4. Relationship between Stress and Positive or Negative Stimulation 

Stress is also susceptible to emotional stimuli. Traditionally, the stress group has been recognized to be affected by negative stimulation, whereas the control group (non-stress group) is affected by positive stimulation [62]. In Figure 8a, we also observed the same results as the stress and control groups derived high LIS values by negative (Session 2) and positive (Session 1) stimulations, respectively. Moreover, as presented in Figure 8b, the eustress group was affected by positive and negative stimulations. In contrast, the distress group was rarely affected by either positive or negative stimulation. Thus, these results allow us to distinguish the eustress and distress groups from the entire stress group.

### 4.5. Analysis of the LIS

With strong exposure to negative stimuli, it is generally accepted that the right frontal hemisphere is more affected according to the hemispheric asymmetry theory [63]. Some researchers have also recently found that applying repetitive transcranial magnetic stimulation (rTMS) to patients suffering from PTSD in the right isometric anterior cerebral cortex positively reduces PTSD symptoms [64,65]. Furthermore, theoretically, both acute and chronic stress can affect different forms of lateralization in the human brain, often resulting in a greater involvement in the right hemisphere [66]. The right hemisphere plays a vital role in cognitive and emotional perception. 

As a result of the comparison of the LIS values in two (i.e., stress vs. control) and three groups (i.e., control vs. eustress vs. distress), the calculated LIS values show right frontal lateralization for both the two and three groups as mentioned in Figure 8. The stress and control groups exhibited a distinctive difference in the LIS values in Session 1, centered on positive images. However, we obtained a large difference in the LIS values between the eustress and distress groups in the positive and negative images (Sessions 1 and 2). The eustress group showed dominant activity in the right hemisphere but not in the distress group. This study observed for the first time that the stress group can be subdivided into eustress and distress groups using the LIS value calculated from the hemodynamic oscillations of fNIRS. 

### 4.6. Experimental Paradigm Using Target Effects

Among the various cognitive tasks that induce stress [67,68], the IAPS stimulus-response task is the most preferred as a relatively reliable task for long durations [69,70]. However, this type of task might not be adequate for stress-related studies of hemodynamic signal attenuation. During the experiment, the participants only looked at a blank screen and the IAPS image on the screen alternately, while their brain signals were recorded [48]. After looking at the images for several seconds, participants took a break by looking at a blank screen (or with a fixation cross) to allow the hemoglobin levels to return to baseline. This procedure should be performed iteratively to collect sufficient brain signal data. However, since this repetition takes a long time to finish because of the inherent hemodynamic delay, the session may be so boring that it can cause participants to feel drowsy, and prevent them from consistently concentrating on the task. Therefore, it can cause weak brain signals. To address this shortcoming, we designed an emotional stimulus-response paradigm to achieve target effects that enhance the participants’ brain signals [8]. To achieve the target effect, all participants were instructed to watch the IAPS images and to distinguish whether they were positive or negative. In this process, the stress groups, including eustress and distress, showed greater negative hemodynamic oscillations in Session 2 (emphasized negative stimuli) (Figure 7). Distinct hemodynamic oscillations were observed between the three groups because stressed people are sensitive to negative emotions [13]. Using this paradigm, we can obtain distinctive differences in the enhanced hemodynamic oscillations between the eustress and distress groups. It is essential to design an appropriate experimental paradigm to subdivide the stress group (Figure 2).

## 5. Conclusions

This study discovered that the LIS values calculated from the hemodynamic oscillations of fNIRS signals were useful for subdividing the stress group into the eustress and distress groups, which was consistent with the sAA test results. This study used a newly designed stimulus-response task using IAPS images to induce positive or negative emotions. This task can strengthen the hemodynamic oscillations of the eustress and distress groups based on the target effects. Through the designed stimulus-response tasks, we found that the eustress group exhibited a dominant activity in the right-front hemisphere, unlike the distress group. We first observed that the stress group could be divided into eustress and distress groups using the LIS value. The results of this study significantly contributed to subdividing the stress group into the eustress and distress groups, compared to previous studies that could only classify stress into stress or non-stress. Moreover, the stress and control groups showed a distinctive difference in the LIS values in Session 1 (centered on a positive image), but the eustress and distress groups showed a distinctive difference in the LIS values in both Sessions 1 and 2 (centered on a negative image). LIS values were high in the order of eustress, control, and distress, regardless of negative or positive external stimuli. Therefore, this study is the first to distinguish between eustress and distress groups from the entire stress group using fNIRS-based LIS. 

## Figures and Tables

**Figure 1 biosensors-12-00033-f001:**
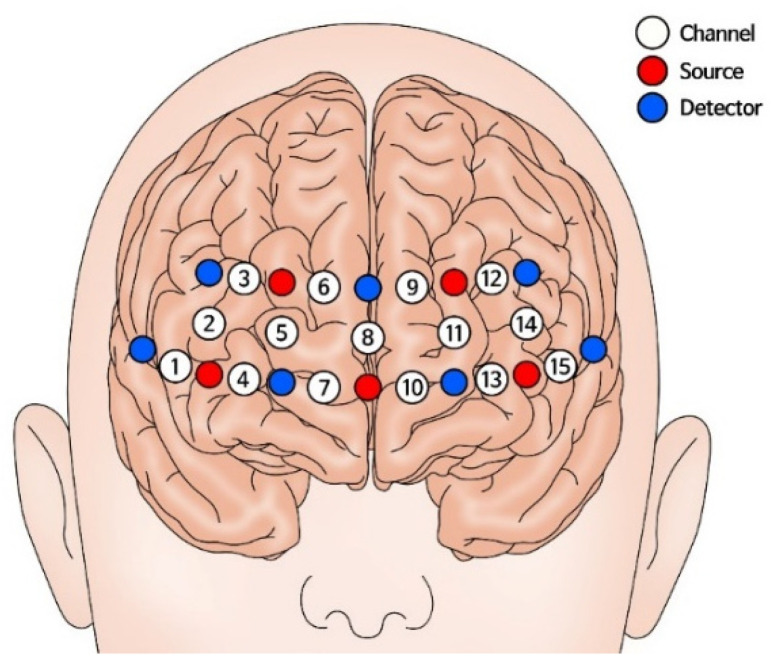
Topographical distribution of the fNIRS recording channels. Twelve probes (5 sources and 7 detectors) were placed, covering the PFC with a probe distance of 25 mm, resulting in a total of 15 channels. The electrodes were positioned at reference point FPz according to the 10/20 international system on each participant’s head.

**Figure 2 biosensors-12-00033-f002:**
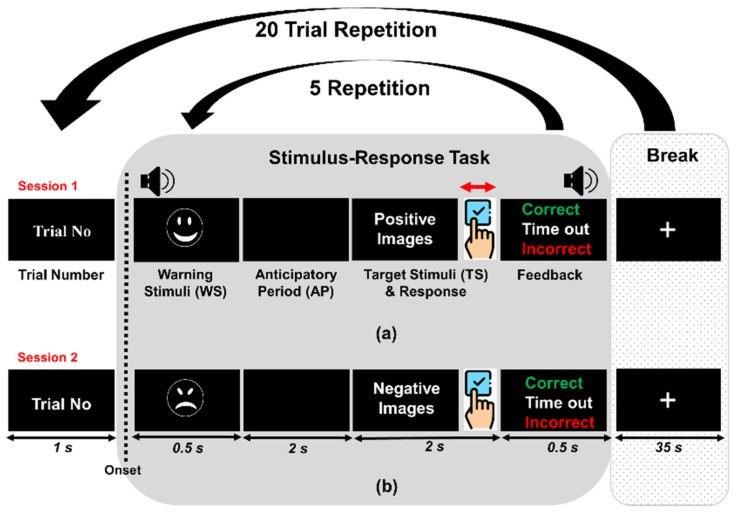
Experimental paradigm. Each participant performed two sessions: (**a**) Session 1 (a positive image ratio (30%) and a negative image ratio (70%)), and (**b**) Session 2 (a negative image ratio (30%) and a positive image ratio (70%)). This experiment compared the hemodynamic responses and behavioral results of each group depending on the ratio of emotional images. In each stage, a custom emotional stimuli-response task was presented for 25 s (i.e., 1 trial: 5 s × 5 repetitions) followed by 35 s rest. A total of 20 trials were conducted.

**Figure 3 biosensors-12-00033-f003:**
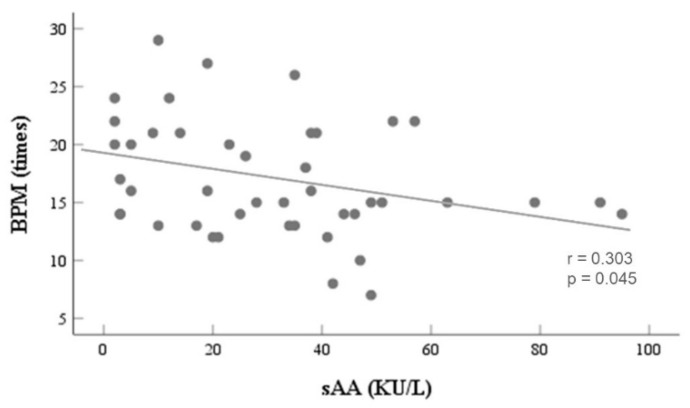
Scatter plots showing the Pearson correlation coefficients between BPM and sAA: The line represents the negative linear trend. The *p*-value and the *r*-value are displayed in the graph. The data from all participants are plotted for these graphs. BPMs are proportionally associated with sAA in all participants.

**Figure 4 biosensors-12-00033-f004:**
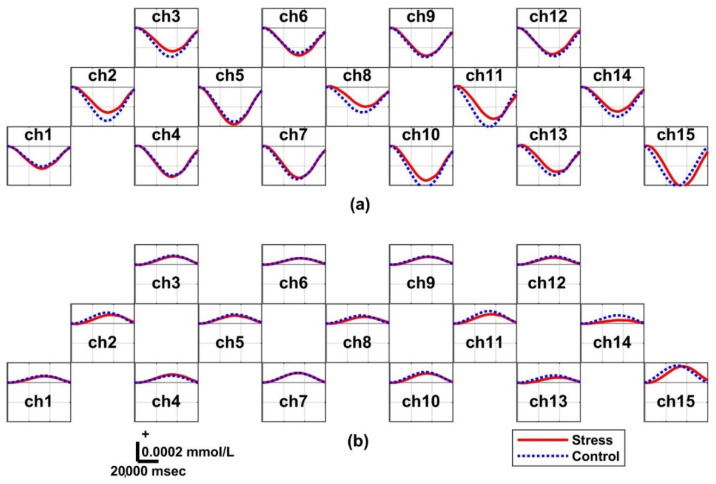
Grand averaged temporal hemodynamic oscillation changes within the interval of each epoch in Session 1 for the control and stress groups: (**a**) oxygenated hemoglobin (ΔHbO) and (**b**) reduced hemoglobin (ΔHbR). The red and blue dotted lines correspond to the stress and control group, respectively. The control group shows slightly higher average brain activation than the stress group under Session 1 of finding positive stimulation conditions.

**Figure 5 biosensors-12-00033-f005:**
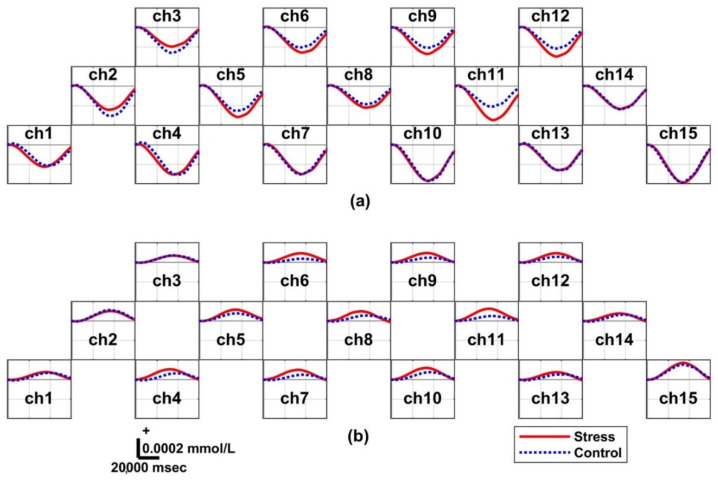
Grand averaged temporal hemodynamic oscillation changes within the interval of each epoch in Session 2 for the control and stress groups: (**a**) oxygenated hemoglobin (ΔHbO) and (**b**) reduced hemoglobin (ΔHbR). The red and blue dotted lines correspond to the stress and control group, respectively. The stress group shows slightly higher brain activation than the control group on average under Session 2 of finding negative stimulation conditions.

**Figure 6 biosensors-12-00033-f006:**
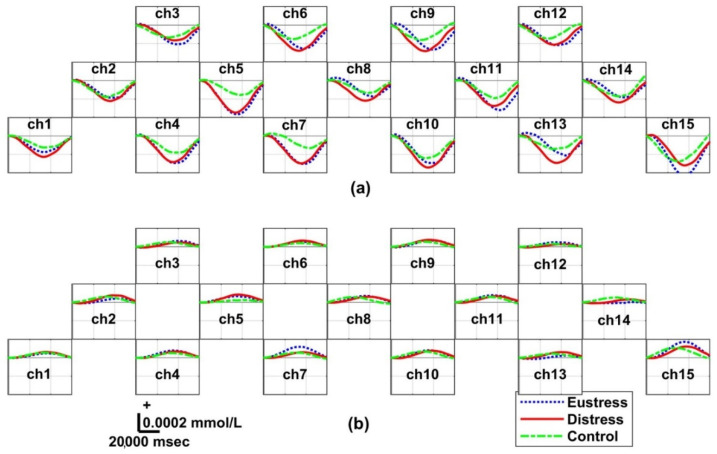
Grand averaged temporal hemodynamic oscillation changes within the interval of each epoch in Session 1 for the eustress, distress, and control groups: (**a**) oxygenated hemoglobin (ΔHbO) and (**b**) reduced hemoglobin (ΔHbR). The blue dotted, red, and green dotted lines correspond to the eustress, distress, and control groups, respectively. These data show the highest brain oscillation in the distress group, followed by the eustress group and control group.

**Figure 7 biosensors-12-00033-f007:**
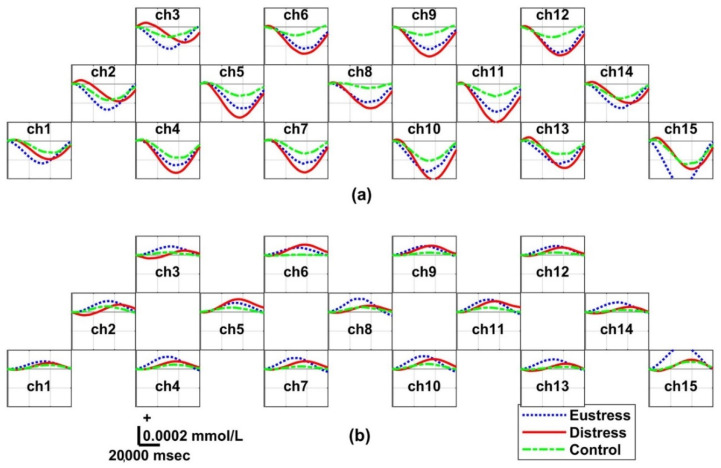
Grand averaged temporal hemodynamic oscillations within the interval of each epoch in Session 2 for the eustress, distress, and control groups: (**a**) oxygenated hemoglobin (ΔHbO) and (**b**) reduced hemoglobin (ΔHbR). The blue dotted, red, and green dotted lines correspond to the eustress, distress, and control groups, respectively. These data show the highest brain oscillation in the distress group, followed by the eustress group and control groups.

**Figure 8 biosensors-12-00033-f008:**
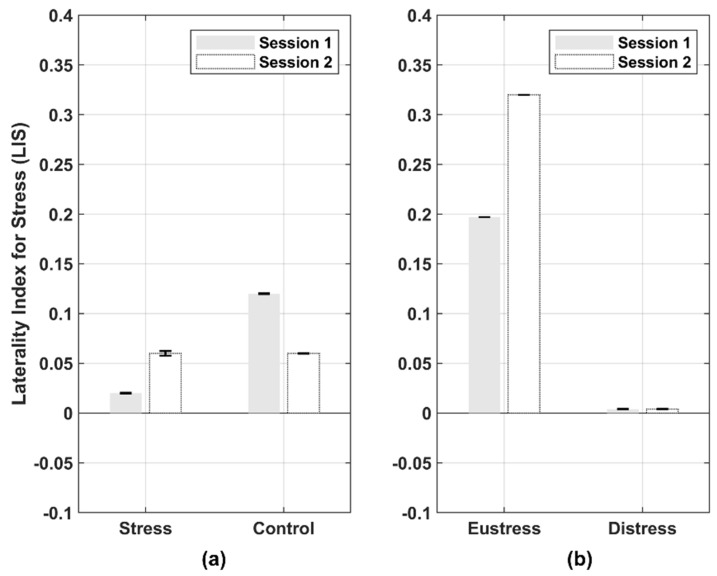
Bar graphs of the averaged laterality index for stress (LIS) between two groups and among three groups by the experimental condition type of Session 1 (showing the target effect by a positive image) and 2 (showing the target effect by a negative image); (**a**) LIS values are right-front biased in the control and stress groups and (**b**) LIS values are also right-front biased in the eustress and distress groups subdivided from stress groups. In (**a**) a large difference in the LIS values between the stress and the control groups is obtained in Session 1 but not in Session 2. However, in (**b**) a large difference in the LIS values between the eustress and the distress groups is obtained in both Session 1 and 2. The calculated LIS values are high in the order of the eustress, control, and distress groups. The error bars refer to a standard error to estimate the variability of the LIS among all the participants in each group.

**Table 1 biosensors-12-00033-t001:** Summary of participants between two groups (n = 44) and three groups (n = 33). All groups were selected depending on the saliva alpha-amylase (sAA) levels. To increase statistical significance, the control group shows subsets of random data (n = 11) chosen three times to have a matched number of samples among the three groups. Note that the physiological results of all groups are observed in normal and similar conditions before the experiment. There were no significant differences in sex and age for each group. However, the sAA level shows obvious statistical significance (** *p* < 0.01).

		Two Groups (n)		*p*-Value ** *p* < 0.01	Three Groups (n)					*p*-Value ** *p* < 0.01
		Control	Stress		Control			Stress		
					Sub1	Sub2	Sub3	Eustress	Distress	
Number of participants		22	22	-	11	11	11	11	11	-
Sex	Male	15	15	1.000	9	6	9	6	6	0.329
	Female	7	7		2	5	2	5	5	
Age, mean and standard deviation (SD)		24.05 ± 2.32	24.41 ± 2.81	0.642	23.73 ± 1.85	23.09 ± 2.55	23.45 ± 2.46	24.91 ± 3.36	23.91 ± 2.17	0.773
sAA (KU/L), mean and standard deviation (SD)		12.73 ± 9.76	49.82 ± 17.67	0.000 **	13.09 ± 10.90	13.27 ± 8.92	14.00 ± 11.29	37.82 ± 3.49	61.82 ± 18.07	0.000 **

**Table 2 biosensors-12-00033-t002:** IAPS image numbers divided into valance and arousal ratings.

Positive Images	Negative Images
1340, 1463, 1710, 2045, 2058, 2071, 2158, 2216, 2340, 2347, 2352.1, 5470, 5480, 5600, 5621, 5623, 5629, 5700, 5814, 5825, 5830, 5833, 5910, 7260, 7330, 7502, 7508, 8030, 8034, 8080, 8090, 8120, 8163, 8170, 8180, 8185, 8186, 8190, 8200, 8210, 8370, 8380, 8400, 8420, 8461, 8470, 8490, 8496, 8499, 8540,	2352.2, 2683, 2688, 2811, 3030, 3103, 3170, 3225, 3266, 3400, 3500, 3530, 3550.1, 6021, 6212, 6230, 6231, 6250.1, 6260, 6300, 6313, 6315, 6350, 6360, 6370, 6415, 6510, 6520, 6530, 6540, 6550, 6560, 6563, 6570.1, 6821, 8485, 9075, 9163, 9183, 9187, 9250, 9300, 9325, 9410, 9412, 9413, 9570, 9635.1, 9921, 9940,
1410, 1440, 1441, 1460, 1540, 1604, 1610, 1630, 1920, 2035, 2040, 2151, 2156, 2165, 2306, 2314, 2331, 2332, 2341, 2388, 2391, 2395, 2550, 2598, 2650, 2660, 5000, 5001, 5200, 5201, 5202, 5210, 5220, 5551, 5594, 5611, 5631, 5725, 5760, 5779, 5780, 5781, 5811, 5829, 5831, 5836, 5891, 5982, 7325, 8497	2053, 2095, 2301, 2345.1, 2375.1, 2456, 2703, 2750, 2751, 2799, 2800, 2900, 3016, 3017, 3160, 3168, 3180, 3181, 3215, 3220, 3230, 3261, 3300, 3301, 3350, 6311, 6831, 9040, 9043, 9140, 9181, 9185, 9220, 9301, 9322, 9326, 9332, 9421, 9423, 9425, 9428, 9429, 9430, 9433, 9560, 9561, 9571, 9610, 9911, 9920

A total of 200 stimuli were chosen from IAPS images, depicting 100 pleasant and 100 unpleasant pictures.

**Table 3 biosensors-12-00033-t003:** Pearson correlation coefficients between stress scales (sAA) and behavioral results (accuracy or response times (RT)): Scale validity of Pearson correlation for two groups (n = 44) and three groups (n = 33). Two groups show a significant positive correlation between sAA and accuracy. However, three groups exhibit a positive correlation between sAA and RT: Session 1. Although the other results do not show statistically significant differences, they suggest another possibility of classifying stress levels between eustress and distress groups.

**Two Groups**	**sAA**	**Accuracy**	**RT: Session 1**	**RT: Session 2**
sAA	1			
Accuracy	0.484 *	1		
RT: Session 1	0.001	−0.369	1	
RT: Session 2	−0.241	−0.104	0.509 *	1
**Three Groups**	**sAA**	**Accuracy**	**RT: Session 1**	**RT: Session 2**
sAA	1			
Accuracy	0.242	1		
RT: Session 1	0.429 *	−0.347	1	
RT: Session 2	0.162	−0.405	0.862 **	1

* *p* < 0.05; ** *p* < 0.01.

**Table 4 biosensors-12-00033-t004:** Comparison of anxiety and stress scales for each group with different sAA levels: Two independent sample *t*-tests and ANOVA (* *p* < 0.05; ** *p* < 0.01).

	Groups	Participants (n)	Mean Score and Standard Deviation (SD)	*t*	*p*-Value * *p* < 0.05; ** *p* < 0.01
STAI-T	Two	Control (22)	41.32 ± 9.27	2.994	0.005 **
		Stress (22)	50.14 ± 10.24		
	Three	Control (11)	41.32 ± 9.27	2.731	0.081
		Eustress (11)	40.82 ± 10.74		
		Distress (11)	49.08 ± 9.80		
STAI-S	Two	Control (22)	39.45 ± 10.22	2.414	0.020 *
		Stress (22)	47.27 ± 11.24		
	Three	Control (11)	41.32 ± 9.27	1.133	0.335
		Eustress (11)	48.00 ± 10.95		
		Distress (11)	46.92 ± 11.52		
PSS	Two	Control (22)	15.91 ± 4.98	2.763	0.008 **
		Stress (22)	20.68 ± 6.39		
	Three	Control (11)	41.32 ± 9.27	2.469	0.102
		Eustress (11)	21.73 ± 5.62		
		Distress (11)	19.25 ± 6.98		

All questionnaires have distinct score differences between the two groups, but no statistically significant differences were found among the three groups. The questionnaire, which has been traditionally and widely used as a stress scale, can not show any possibility of determining precise stress levels using this assessment, even if it can be determined whether the person has experienced stress or not.

## Data Availability

Data of our study are available upon request.

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
