# Peer review of "Subdividing Stress Groups into Eustress and Distress Groups Using Laterality Index Calculated from Brain Hemodynamic Response"

_biosensors, 2022, doi:10.3390/bios12010033_

Round 1

Reviewer 1 Report

In this paper, the authors present the use of fNIRS to measure the brain’s hemodynamic reaction. Compared the two stress subgroups categorized by sAA using emotional stimulus-response paradigm with an international affective picture system to enhance hemodynamic signals induced by the target effect. Then calculated the LIS to distinguish between eustress and distress groups from the entire stress group. This article is clear, concise, and suitable for the scope of the journal. Several small suggestions are supplied:

  1. Suggest giving more detail about the signal processing part.
  2. Suggest give some explain for different of BPM (? = 2.674, ? < .05* ) between the control and stress groups.

Author Response

Point 1: Suggest giving more detail about the signal processing part.

  • We explained it in more detail and added references in section 2.5 (page 6, 1st paragraph, line 198) as follow: “We adopted the signal processing methods used by Al-Shargie, F., et al. [41-42]. Using the modified Beer–Lambert law, changes of oxygenated and reduced oxygenated hemoglobin concentrations converted from fNIRS signals can be represented as and   The converted signals were preprocessed to remove low-frequency drift and high-frequency system noise using the framework of the BBCI-Toolbox (https://github.com/bbci/bbci_public) [43] running in Matlab 2019a (MathWorks, Natick, USA). The signals were digitally bandpass-filtered through a zero-phase filter implemented by a sixth-order Chebyshev filter with a passband of 0.01-0.02 Hz (approximately 1/60 s) to remove physiological noise and DC offset. We obtained the same results using a wide passband of 0.01-0.1 Hz to prevent information loss due to the narrow passband of 0.01-0.02 Hz. The filtered hemodynamic responses were segmented into epochs ranging from -1 to 60 s relative to the task onset (i.e., 0 s). Epochs were subjected to a baseline correction, which consisted of subtracting the average value within the reference interval ranging from -1 to 0 s. The temporal means of  in each channel were calculated by averaging the fNIRS data from the onset (0 s) to the termination time (60 s) of each epoch. Similarly,  was also preprocessed, but  signals better reflect the hemodynamic activation [44]. Thus, we dealt only with  signals in the subsequent analyses [45].”

Point 2: Suggest give some explain for different of BPM (? = 2.674, ? < .05*) between the control and stress groups.

  • We explained it in section 3.3 (page 10, 3rd paragraph, line 341) as follow: “indicating that there is a significant difference in BPM between the two groups.”

Reviewer 2 Report

In this manuscript, the authors investigated the necessity of subdividing the stress group. The stress group was screened by salivary alpha-amylase (sAA) and then, brain’s hemodynamic reactions are measured by functional near-infrared spectroscopy (fNIRS) based on the near-infrared biosensor. Using, a newly designed emotional stimulus-response paradigm with an international affective picture system to enhance hemodynamic signals induced by the target effect. Also, it calculated the laterality index for stress (LIS) from the measured signals to identify the dominantly activated cortex in both subgroups. Both stress groups showed brain activity in the right frontal cortex. Specifically, the eustress group showed the largest brain activity, whereas the distress group showed recessive brain activity, regardless of positive or negative stimuli.

The topic itself is an interesting and timely one. However, the manuscript and the statistical analysis is generally poorly written, which cannot be accepted for publication unless the authors can extensively revise the manuscript under the guidance of a senior research or statistician.

Specifically, 1), there are many grammatical errors throughout the manuscript, which do not allow fluent reading.

2), the correlation results reported in the Table 3 is not clear, however, the authors interpreted the results as if there are significant.

3), Introduction: the authors should first briefly review previous studies of the LIS values obtained from hemodynamic oscillations of fNIRS, emphasize the novelty of the current study, and provide the readers with enough background information.

4), Methods: section 2.10, the authors stated that they investigated correlations between sAA and behavioral results of accuracy or reaction time. Hence, they obtained Pearson correlation coefficients for these data, however, as can be seen from Table 3, this statement was wrong, if the sAA levels none considered corrected, after fMRSI.

5), Results: Figure 4a, 4b, & 5a, 5b, given the characteristics of the data, it seems Pearson correlation cannot be used, please confirm.

Author Response

Point 1: There are many grammatical errors throughout the manuscript, which do not allow fluent reading.

  • Our paper was received English editing servce by ESSAYREVIEW.

Point 2: The correlation results reported in the Table 3 is not clear, however, the authors interpreted the results as if there are significant.

  • We revised the contents related to Table 3. The correlations between sAA and accuracy in two group, and between sAA and response time of session 1 in three group yield statistically significant differences, suggesting that sAA and behavioral results are another possibility to determine stress levels between eustress and distress groups. Thus, we revised the caption of Table 3 (page 8) as follow: “Pearson correlation coefficients between stress scales (sAA) and behavioral results (accuracy or response times (RT)): Scale validity Pearson correlation for two groups (n=44) and three groups (n=33). Two groups show a significant positive correlation between sAA and accuracy. However, three groups exhibit a positive correlation between sAA and RT: Session1. Although the other results do not show statistically, significant differences, they suggest another possibility of classifying stress levels between eustress and distress groups. ”
  • We also supplemented section 3.1 in results (page 8, 2nd paragraph, line 284) as follow: “To reveal a correlation between stress levels (sAA) and behavioral results obtained from the subject’s key-pressing actions, we performed a two-tailed Pearson's correlation analysis, as presented in Table 3. In the two-group analyses, the highest positive correlation was observed between sAA level and accuracy (= .484,  < .05*). On the other hand, in the three-group analyses, the highest positive correlation was observed between sAA level and response time in Session 1 ( = .429,  < .05*). These results indicate that the accuracy and response time obtained from the task are related to the sAA, which provides another possibility to classify stress levels between eustress and distress groups.”

Point 3: Introduction: the authors should first briefly review previous studies of the LIS values obtained from hemodynamic oscillations of fNIRS, emphasize the novelty of the current study, and provide the readers with enough background information.

  • We revised it in Introduction (page 2, 6th paragraph, line 73) as follow: “In many neuroimaging studies, the laterality index (LI) is defined as a hemispheric dominance in mental stress-induced cognitive tasks, such as mental arithmetic, working memory, etc [27]. The laterality index for stress (LIS) is mainly used to calculate an asymmetry coefficient in left/right brain patterns between stressed people and unstressed people [28]. Most studies have shown dominant brain activity in the right hemisphere of people stressed by mental stress tasks [29–31]. Here, the right hemisphere plays an important role in the diagnosis of stress. Furthermore, LI-based studies have reported that high- and low-stress can be evaluated in difficult and easy-task stages, respectively; The more difficult the task to be performed is, the greater is the perceived stress and the dominant activities appear in the right prefrontal cortex (PFC) by LI calculation [32, 33]. However, it has never been reported that the perceived stress is directly related to the physiological stress measured objectively, such as sAA. Moreover, the stress measured by sAA is not yet known whether hemisphere-specific LIS changes influence the PFC in the eustress and distress groups [34]. To address these issues, we divided the actual stress measured by sAA into eustress (low stress) and distress (high stress) and then calculated the LIS values by the hemodynamic oscillations of fNIRS. Furthermore, our research can greatly contribute to measuring the stress lateralization between eustress and distress and differentiate stress groups in detail.”

Point 4: Methods: section 2.10, the authors stated that they investigated correlations between sAA and behavioral results of accuracy or reaction time. Hence, they obtained Pearson correlation coefficients for these data, however, as can be seen from Table 3, this statement was wrong, if the sAA levels none considered corrected, after fMRSI.

  • sAA and behavioral results (accuracy, response times) are not obtained from fNIRS. sAA is defined as saliva alpha-amylase, and the text is modified as a result of the subject's key-pressing action, not the results obtained from fNIRS. Therefore, sAA and behavioral results have nothing to do with fNIRS. Please refer to section 2.8 (page 7, 1st paragraph, line 248) and the caption of Table 3 (page 8) as follow: “Pearson correlation coefficients between stress scales (sAA) and behavioral results (accuracy or response times (RT)): Scale validity Pearson correlation for two groups (n=44) and three groups (n=33). Two groups show a significant positive correlation between sAA and accuracy. However, three groups exhibit a positive correlation between sAA and RT: Session1. Although the other results do not show statistically, significant differences, they suggest another possibility of classifying stress levels between eustress and distress groups. ”

Point 5: Results: Figure 4a, 4b, & 5a, 5b, given the characteristics of the data, it seems Pearson correlation cannot be used, please confirm.

  • We apologize for this error and confusion. We cannot apply Person correlation analyses for Figs. 4a, 4b & 5a, 5b. We revised the mistakes related to Table 3.
